# COVID-19 symptoms predictive of healthcare workers' SARS-CoV-2 PCR results

Fan-Yun Lan[1,2], Robert Filler[1,3], Soni Mathew[3], Jane Buley[3], Eirini Iliaki[3,4], Lou Ann Bruno-Murtha[4], Rebecca Osgood[5,6], Costas A. Christophi[1,7], Alejandro Fernandez-Montero[1,8], Stefanos N. Kales[1,3] *

1 Department of Environmental Health, Harvard University T.H. Chan School of Public Health, Boston, Massachusetts, United States of America, 2 Department of Occupational and Environmental Medicine, National Cheng Kung University Hospital, College of Medicine, National Cheng Kung University, Tainan, Taiwan, 3 Cambridge Health Alliance, Occupational Medicine, Harvard Medical School, Cambridge, Massachusetts, United States of America, 4 Cambridge Health Alliance, Infection Prevention, Infectious Diseases, Harvard Medical School, Cambridge, Massachusetts, United States of America, 5 Cambridge Health Alliance, Pathology, Harvard Medical School, Cambridge, Massachusetts, United States of America, 6 Pathology, Massachusetts General Hospital, Harvard Medical School, Boston, Massachusetts, United States of America, 7 Cyprus International Institute for Environmental and Public Health, Cyprus University of Technology, Limassol, Cyprus, 8 Department of Occupational Medicine, University of Navarra, Pamplona, Spain

* skales@hsph.harvard.edu

**Data Availability Statement:** All relevant data are within the manuscript and its Supporting Information files.

## Abstract

### Background

Coronavirus 2019 disease (COVID-19) is caused by the virus SARS-CoV-2, transmissible both person-to-person and from contaminated surfaces. Early COVID-19 detection among healthcare workers (HCWs) is crucial for protecting patients and the healthcare workforce. Because of limited testing capacity, symptom-based screening may prioritize testing and increase diagnostic accuracy.

### Methods and findings

We performed a retrospective study of HCWs undergoing both COVID-19 telephonic symptom screening and nasopharyngeal SARS-CoV-2 assays during the period, March 9—April 15, 2020. HCWs with negative assays but progressive symptoms were re-tested for SARS-CoV-2. Among 592 HCWs tested, 83 (14%) had an initial positive SARS-CoV-2 assay. Fifty-nine of 61 HCWs (97%) who were asymptomatic or reported only sore throat/nasal congestion had negative SARS-CoV-2 assays (P = 0.006). HCWs reporting three or more symptoms had an increased multivariate-adjusted odds of having positive assays, 1.95 (95% CI: 1.10–3.64), which increased to 2.61 (95% CI: 1.50–4.45) for six or more symptoms. The multivariate-adjusted odds of a positive assay were also increased for HCWs reporting fever and a measured temperature $\geq 37.5°C$ (3.49 (95% CI: 1.95–6.21)), and those with myalgias (1.83 (95% CI: 1.04–3.23)). Anosmia/ageusia (i.e. loss of smell/loss of taste) was reported less frequently (16%) than other symptoms by HCWs with positive assays, but was associated with more than a seven-fold multivariate-adjusted odds of a positive test: OR = 7.21 (95% CI: 2.95–17.67). Of 509 HCWs with initial negative SARS-CoV-2 assays, nine had

**Funding:** The authors received no specific funding for this work.

**Competing interests:** The authors have declared that no competing interests exist.

symptom progression and positive re-tests, yielding an estimated negative predictive value of 98.2% (95% CI: 96.8–99.0%) for the exclusion of clinically relevant COVID-19.

## Conclusions

Symptom and temperature reports are useful screening tools for predicting SARS-CoV-2 assay results in HCWs. Anosmia/ageusia, fever, and myalgia were the strongest independent predictors of positive assays. The absence of symptoms or symptoms limited to nasal congestion/sore throat were associated with negative assays.

## Introduction

Coronavirus 2019 disease (COVID-19) has become pandemic since being first reported in China [1]. COVID-19 can present along a clinical spectrum from asymptomatic, mild symptoms (e.g. cold-like) [2–4], influenza-like (e.g. fever, malaise and myalgia) to severe lower respiratory disease with dyspnea and pneumonia [3–5]. Evidence supports person-to-person transmission, including from asymptomatic patients [6–9]. Severe acute respiratory syndrome coronavirus-2's (SARS-CoV-2) environmental persistence suggests transmission may also result from hand contact with contaminated surfaces [10] followed by facial self-contamination.

Healthcare workers (HCWs) potentially experience greater risks for emerging infectious diseases [11–13] due to occupational exposure to sick patients and virus-contaminated surfaces [14]. Contagious HCWs may infect patients, co-workers and family members. Moreover, the removal of ill HCWs from duty can threaten essential healthcare staffing during an epidemic [15]. Therefore, infection prevention and quick, accurate diagnosis of potential COVID-19 in HCWs are crucial to maintaining hospital operations [16].

Testing HCWs with early/mild symptoms has been approved and prioritized [17,18]. Because they are more likely to be tested, characterization of HCWs' presentations and viral assays provide valuable clinical and epidemiologic perspectives on COVID-19, to compare with hospitalized patients generally representing more severe cases [19]. There is no reference diagnostic test for COVID-19. Currently available tools are symptom reports and a reverse transcriptase polymerase chain reaction (RT-PCR) assay that detects SARS-CoV-2 RNA from naso-/oropharyngeal specimens [20], but whose test characteristics are not established at this time [21]. Because of limited testing capacities, and potential latency of up to 14 days in illness onset from exposure [22], HCW testing is largely restricted to persons reporting compatible symptoms [17,18]. Nonetheless, many HCWs and other first responders desire wider testing because of potential exposures, often without symptoms. Therefore, we investigated the presenting symptoms most predictive of positive/negative SARS-CoV-2 RT-PCR results among HCWs.

## Methods

### Study population and setting

Since March 9, 2020, the occupational health service of a Massachusetts community healthcare system has implemented a staff "hotline" system to maintain a viable/healthy workforce and operational continuity during the pandemic. Accordingly, the service was re-configured to perform telephonic triage of HCWs for COVID-19-related concerns; electronically document

related clinical information; manage and communicate pandemic-related testing results; and oversee HCWs' safe return to work.

Reasons for contacting the COVID-19 hotline have included: a) travel; b) potential contact with a COVID-19-positive/suspect person; and/or c) possible viral symptoms. A standard triage form (S1 File) was completed by occupational nursing and medical personnel based on contemporaneous telephonic interviews with each HCW. The form contains demographics, administrative information, potential exposure history, and eleven potential viral symptoms: fever (subjective, as well as highest measured temperature), cough (new or worse), shortness of breath (new or worse), myalgia, malaise, sore throat, nasal symptoms (including runny nose, sneezing, congestion, and sinus symptoms), gastrointestinal symptoms (including nausea, vomiting, and diarrhea), rash, anosmia/ageusia (i.e. loss of smell/loss of taste), and headache. Headache and anosmia/ageusia were added to a revised form after several HCW reports.

In accordance with expert guidelines [17,18], symptomatic HCWs were referred for SARS-CoV-2 testing (see below). Employees tested elsewhere forwarded their RT-PCR results to the occupational health service and were triaged via a telephonic visit. In this retrospective cohort study, we included all adult employees/personnel or contractors of the hospital who had undergone both COVID-19-related triage and a SARS-CoV-2 PCR assay between March 9 and April 15, 2020.

## Specimen collection and testing

SARS-CoV-2 assays were performed at designated sites, where trained healthcare system staff collected nasopharyngeal swabs and transferred each specimen to a 3ml vial with viral transport media (VTM), Universal Transport Media (UTM) or saline. After collection, specimens were refrigerated at 2–8˚C, unless transported immediately, and refrigerated at 2–8˚C during transport. HCWs samples were sent to one of three laboratories (the Massachusetts Department of Public Health (MADPH), a commercial lab, or a tertiary hospital partner), whichever offered the fastest turnaround-time on the day of testing. All three laboratories used real-time, reverse-transcriptase–polymerase-chain-reaction (RT-PCR) diagnostic panels for the qualitative detection of nucleic acid from the 2019 SARS-CoV-2 (MADPH, CDC 2019-Novel RT-PCR; commercial laboratory, Roche Cobas SARS-CoV-2; and hospital partner, Abbott Real Time SARS-CoV-2). The limit of detection at the laboratories testing the majority of our samples was 100 copies of viral RNA/ml. Positive assay results represented detection of SARS-CoV2 RNA, while for negative results, the virus was not detected. In case of an invalid or indeterminate result, a repeat sample is requested.

## Data collection

For operational needs, the occupational health service maintains data from the triage encounters, PCR results and clinical/work status on a secure "live" spreadsheet. The presence/absence of the eleven possible symptoms, travel, and exposure history in the spreadsheet entries were verified from triage forms by an occupational medicine physician for every HCW undergoing a RT-PCR assay. For body temperature, we recorded the highest value before testing as self-reported by each HCW. SARS-CoV-2 PCR assay results for HCWs tested outside of the healthcare system were verified and entered in the database.

Because there is no reference diagnostic test, we considered false negative SARS-CoV-2 assays as occurring when the initial test was negative, but symptoms had progressed/persisted at telephonic follow-up and a repeat assay was subsequently SARS-CoV-2 positive.

### Human subjects

The Institutional Review Board of the healthcare system reviewed the study protocol, determined it to be exempt and waived informed consent based on the use of existing, HIPAA-deidentified data. Prior to statistical analyses, all data were deidentified and transferred to a spreadsheet without linkages to the "live" database.

### Statistical analysis

Continuous characteristics were presented as means and standard deviations and compared between groups with the parametric t-test. Categorical variables were presented as counts and percentages and compared between groups using the chi-square test of independence with the Yates' continuity correction or the Fisher's exact test, as appropriate. No imputations were made for missing data.

Principal component analysis (PCA) was utilized to identify dominant symptoms (those contributing more than 5% variability to the identified principal component explaining most data variability). Logistic regression models were then fit to evaluate symptom association with the probability of having a positive SARS-CoV-2 assay, after age-, sex- and other symptom adjustment. Regression results are provided as odds ratios (ORs) and corresponding 95% confidence intervals (CIs). The Hosmer-Lemeshow goodness of fit test with 15 groups was also performed to check the model fit. Additionally, the C-statistic was calculated to assess each model's predictive accuracy.

The clinical COVID-19 attack rate during the study period was calculated as: (the number of initial positive SARS-CoV-2 assays + the number of false negatives) divided by the system's estimated total HCW population (n = 4600). COVID-19 complication rates were calculated as HCWs who required an emergency room visit or hospitalization before the end of the study period (April 15, 2020) and/or had intubation or death before the end of follow-up (April 20, 2020); divided by the number of total COVID-19-diagnosed HCWs.

Statistical analyses were conducted using the R software (version 3.6.3). All tests were two-sided and a *P*-value < 0.05 was considered statistically significant.

## Results

### Demographic and clinical characteristics

During the study period, we identified 592 unique HCWs who underwent triage and SARS-CoV-2 RT-PCR. Eighty-three (14.0%) had positive SARS-CoV-2 RT-PCR results on the initial assay. The cohort's presentation at triage is summarized in Table 1. The average age was 43.6 ± 12.9 years old, with no significant difference between HCWs testing positive or negative (*P* = 0.84). The proportion of men with positive tests (27.7%) was non-significantly higher than among those with negative tests (20.0%) (*P* = 0.15).

SARS-CoV-2 positive HCWs self-reported several symptoms more frequently than those with negative assays: fever (55% vs. 27%), myalgia (57% vs. 35%), headache (41% vs. 28%), and anosmia/ageusia (16% vs. 3%) (all *P*<0.05). Nasal symptoms (runny, sneezing, congestion, sinus) were more frequently associated with negative assays (52% vs. 35%) (*P* = 0.006) (Table 1, Fig 1).

Among HCWs reporting fever, 87% (40/46) of those testing SARS-CoV-2 positive and 89% (121/136) of those testing negative had measured their body temperature. The mean peak temperature reported was higher in HCWs with a positive assay (38.0 ± 0.7˚C) compared to those with negative assays (37.6 ± 0.7˚C) (*P* = 0.006). Dichotomizing the peak temperature

**Table 1. Symptom and body temperature distributions at time of triage among healthcare workers (HCWs) by SARS-CoV-2 test results.**

|  | Overall (N = 592) | Positive (N = 83) | Negative (N = 509) | P value |
|---|---|---|---|---|
| Age | 43.6 (12.9) | 43.9 (12.7) | 43.6 (12.9) | 0.843 |
| Female | 467 (78.9%) | 60 (72.3%) | 407 (80.0%) | 0.149 |
| Fever | 182 (30.7%) | 46 (55.4%) | 136 (26.7%) | **<0.001** |
| Measured Temperature (˚C) | 37.68 (0.71) (n = 161) | 37.95 (0.69) (n = 40) | 37.60 (0.69) (n = 121) | **0.006** |
| Temperature ≥ 37.5˚C | 102 (63.4%) (102/161) | 34 (85.0%) (34/40) | 68 (56.2%) (68/121) | **0.002** |
| Cough | 365 (61.7%) | 59 (71.1%) | 306 (60.1%) | 0.074 |
| Shortness of breath | 111 (18.8%) | 14 (16.9%) | 97 (19.1%) | 0.747 |
| Myalgia | 225 (38.0%) | 47 (56.6%) | 178 (35.0%) | **<0.001** |
| Malaise | 274 (46.3%) | 47 (56.6%) | 227 (44.6%) | 0.055 |
| Sore throat | 320 (54.1%) | 38 (45.8%) | 282 (55.4%) | 0.131 |
| Nasal symptoms (runny, sneezing, congestion, sinus) | 293 (49.5%) | 29 (34.9%) | 264 (51.9%) | **0.006** |
| Gastrointestinal symptoms (nausea/ vomiting/ diarrhea) | 151 (25.5%) | 20 (24.1%) | 131 (25.7%) | 0.856 |
| Rash | 10 (1.7%) | 3 (3.6%) | 7 (1.4%) | 0.154[a] |
| Anosmia/Ageusia | 27 (4.6%) | 13 (15.7%) | 14 (2.8%) | **<0.001** |
| Headache | 175 (29.6%) | 34 (41.0%) | 141 (27.7%) | **0.020** |

Mean (SD) for age and body temperature, count (percentage) for all other variables.

[a] Based on Fisher's exact test.

(≥ or <37.5˚C), the measured temperature exceeded the threshold in 85% of HCWs with positive RT-PCR, compared to 56% of those with negative assays (*P* = 0.002).

## Symptoms and probabilities of positive SARS-CoV-2 assay

Assay results for asymptomatic or mildly symptomatic HCWs are shown in Table 2. None of the HCWs with only sore throat and/or nasal symptoms had a positive SARS-CoV-2 assay (0%), while all 34 (100%) had a negative PCR (*P* = 0.009). When we combined asymptomatic and mildly symptomatic HCWs, 59/61 (97%) had negative initial assays (*P* = 0.006).

Total symptoms at triage ranged from zero to ten, with only 40 (7%) HCWs reporting seven or more symptoms. Table 3 shows the counts and percentages of HCWs with positive and negative assays, and age- and sex-adjusted odds ratios for increasing numbers of total reported symptoms. HCWs workers reporting three or more symptoms had an increased likelihood of a positive SARS-CoV-2 assay (OR = 1.95 (95% CI: 1.10–3.64). The odds ratio of a positive RT-PCR generally increased with additional symptoms and reached 2.61 (95% CI: 1.50–4.45) for six or more symptoms.

Further multivariate logistic regression analyses of PCA-determined dominant symptoms are shown in Table 4. HCWs reporting fever had an age- and sex-adjusted odds ratio of 3.34 (95% CI: 2.07–5.41) of having a positive SARS-CoV-2 assay, which remained significant after additional adjustment for other symptoms (OR = 2.88, 95% CI: 1.66–5.01). When measured body temperature ≥37.5˚C was considered together with reported fever, the odds of a positive SARS-CoV-2 assay increased to 4.47 (95% CI: 2.66–7.48) in the age- and sex-adjusted model and to 3.49 (95% CI: 1.95–6.21) in the all-symptom-adjusted model. The odds ratios remained similar when the body temperature threshold changed to 38˚C (OR = 4.45, 95% CI: 2.30–8.44 in the age- and sex-adjusted model; OR = 2.85, 95% CI: 1.36–5.86 in the full model).

HCWs reporting anosmia/ageusia had an increased age- and sex- adjusted odds ratio of 6.50 (95% CI: 2.89–14.51), and multivariate odds ratio of 7.21 (95% CI: 2.95–17.67) of having a positive SARS-CoV-2 assay. Myalgia was also associated with increased odds ratios of having a

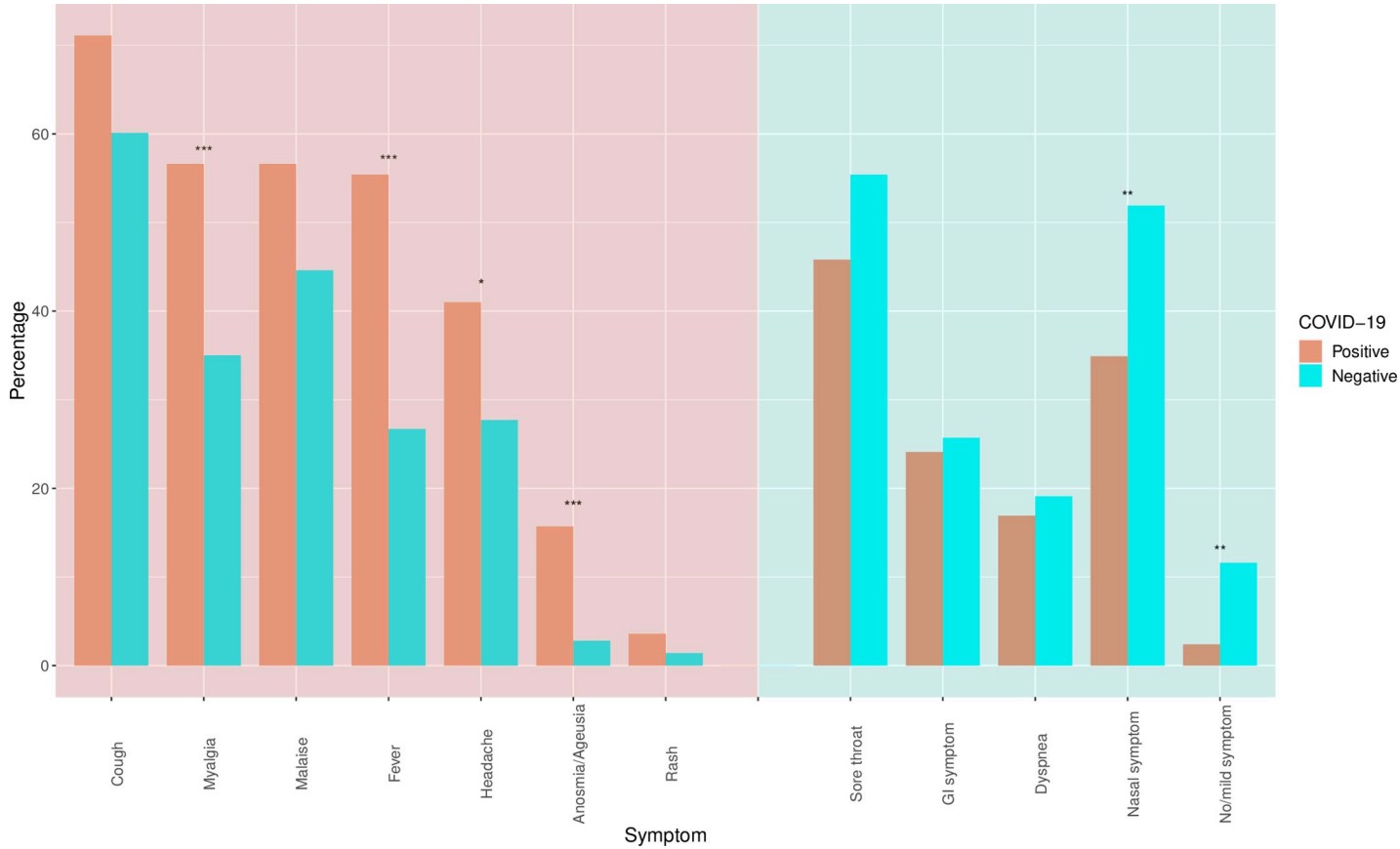

**Fig 1. Symptom distributions among HCWs with initial SARS-CoV-2 (the virus causing COVID-19) testing results.** The symptoms shaded in salmon-red are the symptoms more frequently seen with positive tests, and the symptoms shaded in blue-green are more frequently seen with negative tests. GI symptom denotes a gastrointestinal symptom (nausea/ vomiting/ diarrhea). Nasal symptom includes runny nose, sneezing, congestion, and sinus symptoms. No/mild symptom denotes no symptom or only sore throat and/or nasal symptoms. The asterisks above the bars denote different statistically significance levels when comparing HCWs with positive assays and HCWs with negative assays (*: P<0.05, **: P<0.01, ***: P<0.001).

positive SARS-CoV-2 assay in both models: OR = 2.41 (95% CI: 1.50–3.89) and OR = 1.83 (95% CI: 1.04–3.23), respectively. Headache was a significant predictor in the age- and sex-adjusted model: OR = 1.84 (95% CI: 1.13–2.96), but not in the all-symptom-adjusted model.

Nasal congestive symptoms were a significant negative predictor in both multivariate models: OR = 0.51 (95% CI: 0.31–0.82) and OR = 0.40 (95% CI: 0.23–0.68), respectively.

**Table 2. SARS-CoV-2 test results in asymptomatic healthcare workers (HCWs) and HCWs with only nasal/throat symptoms.**

|  | Positive (N = 83) | Negative (N = 509) | P value [a] |
|---|---|---|---|
| No symptom | 2 (2.4%) | 25 (4.9%) | 0.406 |
| Only sore throat | 0 (0%) | 12 (2.4%) | 0.390 |
| Only nasal symptoms | 0 (0%) | 7 (1.4%) | 0.601 |
| Only sore throat and/or nasal symptoms | 0 (0%) | 34 (6.7%) | **0.009** |
| No symptom or Only sore throat and/or nasal symptoms | 2 (2.4%) | 59 (11.6%) | **0.006** |

[a] Fisher's exact test

**Table 3. SARS-CoV-2 test results by number of symptoms reported at triage [a].**

|  | Positive (N = 83) | Negative (N = 509) | Age and sex adjusted OR (95% CI) |
|---|---|---|---|
| $<$ 2 symptoms | 7 (8.4%) | 71 (13.9%) | 0.57 (0.23–1.20) |
| $\geq$ 2 symptoms | 76 (91.6%) | 438 (86.1%) | 1.77 (0.83–4.36) |
| $\geq$ 3 symptoms | 68 (81.9%) | 357 (70.1%) | **1.95 (1.10–3.64)** |
| $\geq$ 4 symptoms | 53 (63.9%) | 238 (46.8%) | **2.00 (1.24–3.28)** |
| $\geq$ 5 symptoms | 35 (42.2%) | 152 (29.9%) | **1.72 (1.06–2.77)** |
| $\geq$ 6 symptoms | 24 (28.9%) | 69 (13.6%) | **2.61 (1.50–4.45)** |

Abbreviations: OR, odds ratio; CI, confidence interval.

[a] The whole list of symptoms reported were fever, cough, shortness of breath, myalgia, malaise, sore throat, nasal symptoms (runny, sneezing, congestion, sinus), gastrointestinal symptoms (nausea/ vomiting/ diarrhea), rash, anosmia/ageusia (i.e. loss of smell/loss of taste), and headache.

The Hosmer-Lemeshow goodness of fit test for all models demonstrated no evidence of poor fit. Therefore, we also present C statistics (i.e. area under the curve) for each symptom in Table 4. After age- and sex-adjustment, fever had the best discrimination between HCWs with positive and negative SARS-CoV-2 assays (C-statistic = 0.663).

## Estimated false negatives, negative predictive value, attack and complication rates

Nine HCWs with symptom progression after an initial negative SARS-CoV-2 assay are described in S1 Table. Upon re-testing, all nine had positive RT-PCR for SARS-CoV-2. These cases represent 1.8% of originally negative HCWs (9/509), yielding an estimated negative predictive value of 98.2% (95% CI: 96.8–99.0%) for excluding clinically relevant COVID-19 disease.

The cumulative attack rate for clinically symptomatic COVID-19 during the study period (March 9-April 15, 2020) was 92 incident cases among an estimated employee cohort of 4600

**Table 4. Odds ratios of a positive SARS-CoV-2 assay by triage symptom.**

|  | Positive (N = 83) | Negative (N = 509) | Age and sex adjusted OR (95% CI) | Area under ROC curve | Multivariate Adjusted OR (95% CI) [b] |
|---|---|---|---|---|---|
| Fever | 46 (55.4%) | 136 (26.7%) | **3.34 (2.07–5.41)** | 0.663 | **2.88 (1.66–5.01)** |
| Fever plus temperature $\geq$ 37.5˚C [a] | 33 (39.8%) | 65 (12.8%) | **4.47 (2.66–7.48)** | 0.660 | **3.49 (1.95–6.21)** [c] |
| Myalgia | 47 (56.6%) | 178 (35.0%) | **2.41 (1.50–3.89)** | 0.625 | **1.83 (1.04–3.23)** |
| Malaise | 47 (56.6%) | 227 (44.6%) | 1.58 (0.99–2.55) | 0.575 | 1.08 (0.60–1.92) |
| Shortness of breath | 14 (16.9%) | 97 (19.1%) | 0.85 (0.44–1.54) | 0.539 | 0.66 (0.32–1.28) |
| Nasal symptoms (runny, sneezing, congestion, sinus) | 29 (34.9%) | 264 (51.9%) | **0.51 (0.31–0.82)** | 0.601 | **0.40 (0.23–0.68)** |
| Gastrointestinal symptoms (nausea/ vomiting/ diarrhea) | 20 (24.1%) | 131 (25.7%) | 0.91 (0.52–1.54) | 0.538 | 0.54 (0.28–1.00) |
| Headache | 34 (41.0%) | 141 (27.7%) | **1.84 (1.13–2.96)** | 0.572 | 1.43 (0.82–2.47) |
| Anosmia/ageusia | 13 (15.7%) | 14 (2.8%) | **6.50 (2.89–14.51)** | 0.588 | **7.21 (2.95–17.67)** |

Abbreviations: OR, odds ratio; CI, confidence interval; ROC curve, receiver operating characteristic curve.

[a] HCWs not measuring their temperature were assumed to not have elevated body temperature as a sensitivity analysis

[b] Adjusted for age, sex, and all symptoms, i.e. fever, cough, shortness of breath, myalgia, malaise, sore throat, nasal symptom (runny, sneezing, congestion, sinus), gastrointestinal symptom (nausea/ vomiting/ diarrhea), rash, anosmia/ageusia (i.e. loss of smell/loss of taste), and headache

[c] Adjusted for all variables as above except fever

or 2.0% (95% CI: 1.6–2.5%). A total of nine among the 92 incident HCW COVID-19 cases (9.8%) (all nine diagnosed with the initial PCR assay) experienced a complication during the study period: four required an emergency room visit, five needed hospitalization, two of the hospitalized required intubation, and one of those died (1.1%).

## Discussion

This original investigation describes initial symptoms and their associations with SARS-CoV-2 RT-PCR assays among all 592 HCWs in our system tested between March 9 and April 15, 2020. Overall, we tested 13% of our workforce, which is a testing rate 6.7-fold greater than that of the Massachusetts population during the same period [23]. We found that a total of 16% (including 14% with initial positive assays and 2% with initial false negative assays) of these HCWs were diagnosed with clinical COVID-19, yielding a cumulative attack rate of 2% for our workforce.

The absence of symptoms or those limited to the throat/nose (excluding anosmia/ageusia) were significantly associated with having negative SARS-CoV-2 assays. In contrast, HCWs reporting three or more symptoms had 2-fold greater age- and sex-adjusted odds of having positive assays. These odds generally increased for each additional symptom reported, reaching 2.6-fold for six or more symptoms. Our results are consistent with previous reports suggesting that RT-PCR results correlate with viral shedding, COVID-19 symptom onset and clinical severity [24,25]. In multivariate adjustment models, HCWs with anosmia/ageusia had higher than seven-fold odds of having positive RT-PCR; those with fever and a measured temperature $\geq$ 37.5˚C had almost 3.5-fold odds of having positive RT-PCR; and those with myalgia had almost 2-fold odds of having positive assays, while nasal symptoms were associated with a 60% reduced risk of a positive assay.

Studies support varying clinical manifestations among COVID-19 patients [2–5]. An investigation of critically ill cases found half presenting with temperature $\geq$ 38˚C during their ICU stay, while more than 80% had cough and/or shortness of breath, accompanied by tachypnea [19]. In contrast, a study of subclinical COVID-19 patients showed 40% of cases initially presented fever and non-specific symptoms, such as cough and sore throat, but none exhibited dyspnea [4]. Regarding HCWs, it is useful to compare our results with two recent HCW series, which reported average ages (42–43 years-old) and female predominance (73–77%) similar to ours. In the large US national series, 78% of HCWs with COVID-19 had cough, 68% had fever, 66% reported myalgia, and 41% presented shortness of breath [26]. In the second smaller series from King County, WA (USA), the most common initial symptoms were cough (50%), fever (42%) and myalgias (35%) [27]. In general agreement, our HCW COVID-19 cases commonly reported fever (55%), myalgia (57%) and cough (71%), but only 17% reported dyspnea. However, only our investigation collected data on those testing negative as well, demonstrating that cough is a very non-specific symptom that was also reported by 60% of those testing negative. On the other hand, fever (particularly if measured) and myalgia were independent predictors of positive SARS-CoV-2 assays, as were anosmia/ageusia in HCWs symptomatic for longer periods. Regarding the latter findings, they are in accordance with a cross-sectional study of the general population, which reported adjusted odds ratios for COVID-19 of 6.6 and 3.1 for anosmia/ageusia and fever, respectively [28], similar to our findings of fully-adjusted odds of 7.2 and 3.5 for the same symptoms. In our study, anosmia/ageusia was reported less frequently (16%) as compared to 59% in the cross-sectional study, most likely because our symptoms reports were captured early in the course of illness.

Our results are also comparable to a Dutch study examining the point prevalence of positive SARS-CoV-2 RT-PCRs among HCWs with mild respiratory symptoms, during the early

outbreak in the Netherlands, finding 4% of HCWs across nine hospitals positive, ranging from 0–10% for each individual hospital [14]. We found an overall positive initial test rate of 14% during a five-week period, which overlapped early spread with a later surge in community transmission and hospital COVID-19 traffic. We have observed that the proportion of positive tests increased along with epidemic's progression in our region.

False negative results of SARS-CoV-2 RT-PCR have been reported at rates from 10–30% [21,29]. We did find nine HCWs who initially tested negative, developed progressive symptoms and had subsequent positive RT-PCR. None of these likely false-negatives were among the HCWs with the most severe cases requiring emergency/hospital care. Further study of symptomatic patients, their household contacts and of random population samples with both RT-PCR and serologies [30] are needed to better elucidate the true prevalence of asymptomatic carriers as well as the frequency of false negative RT-PCRs. Nonetheless, our estimated negative predictive value of 98% for excluding clinically relevant COVID-19 disease is very reassuring regarding the test's performance when done properly.

Our findings have potential implications for HCW COVID-19 surveillance. First, HCWs with no symptoms or mild symptoms limited to sore throat/nasal congestion and a measured body temperature lower than 37.5°C, have a low probability of a positive RT-PCR and COVID-19. Thus, many HCWs with no symptoms or symptoms more compatible with allergy or a common cold could refrain from testing and continue to self-monitor symptoms and body temperature. In contrast, early systemic symptoms, especially fever and myalgia are predictive of possible clinical COVID-19, while anosmia/ageusia are fairly specific later findings. Our results support expert guidelines for temperature monitoring [18], which should not be limited to healthcare settings, and that a measured temperature $\geq$ 37.5°C may be predictive of a positive SARS-CoV-2 RT-PCR result.

Our study does have some limitations. First, the HCWs tested were not a random or convenience sample, but rather all HCWs who self-reported COVID-19-related concerns. Thus, our results do not necessarily reflect total cumulative incidence of COVID-19 among HCWs in our healthcare system. However, our triage and testing process were conducted in accordance with expert guidelines [3,17,18], and therefore, our results can inform other healthcare systems employing comparable protocols. Second, because there is no reference diagnostic test for COVID-19, the test performance characteristics of the RT-PCR are not established [21]. Potentially corroborating serology tests were not available during the study period, but may be helpful in the future to identify additional infections that occurred with mild or no symptoms [30].

Our study also has several strengths. First, HCWs generally reported their symptoms at triage before their SARS-CoV-2 PCR test results were available, eliminating recall bias. Second, all symptoms reports were validated by occupational medicine physicians, strengthening their accuracy. Third, all HCWs had complete symptom and testing data. Moreover, unlike other HCW studies, we used identical methods to collect data on persons testing negative and were able to compare these two groups. In addition, all HCWs tested had telephonic follow-up visits, allowing us to identify and re-test HCWs with potential false negative results. Finally, our present study included all eligible employees in the healthcare system. The study population comprised not only healthcare professionals but also staff/contractors such as maintenance or IT (Information Technology) persons. Therefore, our results may be generalized to other working populations during the pandemic.

## Conclusions

Among HCWs, systemic symptoms/signs (fever, body temperature $\geq$ 37.5°C, myalgia, and headache) and anosmia/ageusia were predictive of positive SARS-CoV-2 assays, with fever,

anosmia/ageusia, and myalgia being the strongest independent predictor. In contrast, no symptoms or isolated sore throat/nasal congestion were associated with negative SARS-CoV-2 assays.

## Supporting information

**S1 File. The healthcare system's occupational health COVID-19/Influenza like illness triage form.**
(PDF)

**S1 Table. Presentation of HCWs with initial negative assays but repeat positive assays (i.e. presumed initial false negative assays).**
(DOCX)

## Author Contributions

**Conceptualization:** Lou Ann Bruno-Murtha, Rebecca Osgood, Stefanos N. Kales.

**Data curation:** Robert Filler, Soni Mathew, Jane Buley, Eirini Iliaki, Rebecca Osgood, Stefanos N. Kales.

**Formal analysis:** Fan-Yun Lan, Costas A. Christophi, Stefanos N. Kales.

**Funding acquisition:** Stefanos N. Kales.

**Investigation:** Robert Filler, Soni Mathew, Jane Buley, Eirini Iliaki, Stefanos N. Kales.

**Methodology:** Lou Ann Bruno-Murtha, Costas A. Christophi, Stefanos N. Kales.

**Project administration:** Stefanos N. Kales.

**Resources:** Robert Filler, Stefanos N. Kales.

**Software:** Fan-Yun Lan, Costas A. Christophi, Stefanos N. Kales.

**Supervision:** Stefanos N. Kales.

**Validation:** Fan-Yun Lan, Robert Filler, Soni Mathew, Jane Buley, Eirini Iliaki, Lou Ann Bruno-Murtha, Rebecca Osgood, Costas A. Christophi, Alejandro Fernandez-Montero, Stefanos N. Kales.

**Visualization:** Fan-Yun Lan, Stefanos N. Kales.

**Writing – original draft:** Fan-Yun Lan, Costas A. Christophi, Stefanos N. Kales.

**Writing – review & editing:** Fan-Yun Lan, Robert Filler, Soni Mathew, Jane Buley, Eirini Iliaki, Lou Ann Bruno-Murtha, Rebecca Osgood, Costas A. Christophi, Alejandro Fernandez-Montero, Stefanos N. Kales.

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
