## [Decision Letter · Decision Letter 0]

10 Jun 2020

PONE-D-20-13207

COVID-19 Symptoms Predictive of Healthcare Workers’ SARS-CoV-2 PCR Results

PLOS ONE

Dear Dr. Stefanos Kales,

Thank you for submitting your manuscript to PLOS ONE. After careful consideration, we feel that it has merit but does not fully meet PLOS ONE’s publication criteria as it currently stands. Therefore, we invite you to submit a revised version of the manuscript that addresses the points raised during the review process.

I have received the comments of the reviewers on your manuscript. The specific comments of the reviewers are included below. Please provide point by point response in your revised manuscript.

We look forward to receiving your revised manuscript.

Kind regards,

Muhammad Adrish

Academic Editor

PLOS ONE

Journal Requirements:

2. Please include additional information regarding the survey or questionnaire used in the study and ensure that you have provided sufficient details that others could replicate the analyses.

For instance, if you developed a questionnaire as part of this study and it is not under a copyright more restrictive than CC-BY, please include a copy, in both the original language and English, as Supporting Information.

Reviewers' comments:

Reviewer's Responses to Questions

**Comments to the Author**

1. Is the manuscript technically sound, and do the data support the conclusions?

Reviewer #1: Yes

Reviewer #2: Yes

Reviewer #3: Yes

2. Has the statistical analysis been performed appropriately and rigorously? 

Reviewer #1: Yes

Reviewer #2: Yes

Reviewer #3: Yes

3. Have the authors made all data underlying the findings in their manuscript fully available?

Reviewer #1: Yes

Reviewer #2: Yes

Reviewer #3: Yes

4. Is the manuscript presented in an intelligible fashion and written in standard English?

Reviewer #1: Yes

Reviewer #2: Yes

Reviewer #3: Yes

5. Review Comments to the Author

Reviewer #1: Dr. Lan et al. has presented a retrospective observational study in which they investigated the presenting symptoms most predictive of positive/negative SARS-CoV-2 RT-PCR results among healthcare workers. Although the manuscript is well written and the methods are adequate, the observations made have already been described in the general population. HCWs should not be an exception.

Reviewer #2: This manuscript concerns a study among health care workers (HCW) and factors that are associated with testing positive for SARS-2 coronavirus. The data are sound and interesting and may be helpful to determine efficient testing strategies.

Some points need to be considered and adjusted before this paper is suitable for publication in PlosOne.

General comments:

-The Discussion section is rather long and may be shortened by removing redundant statements. For example page 22, second paragraph: this overlaps with other statements in the Discussion.

-In the conclusions of the abstract and at the end of the manuscript, is stated that anosmia/ageusia and fever were the two strongest predictors of positive assays. In Table 1 and Table 4 however, it is clear that also myalgia is significantly associated with positive SARS-2 test results. This is also mentioned in the discussion. Why is this not mentioned in the conclusions in both abstract and discussion?

-In addition: please mention as early as possible in the manuscript that ‘anosmia/ageusia’ means ‘smell/taste’ since these are medical terms with which not all readers are familiar.

-Could the authors speculate on the usefulness of their data for the general population? So would these indicators for a positive SARS-2 test also be true for non-HCW?

Specific comments

Abstract

-Page 3, 5th line: please insert ‘was’ between ’but’ and ‘associated’.

Methods

-Page 7/8: here the RT-PCR assays are mentioned and it is stated that results were either ‘positive’ or ‘negative’. Does this mean that there were no ‘indeterminate’ results? That is a bit hard to believe. If there were indeed ‘indeterminate’ results, how were these dealt with?

-Page 8, Data collection, second paragraph: please replace ‘Absence of a gold standard’ by ‘Because there is no reference diagnostic test’, since the term ‘gold standard’ is not appropriate. The same point for page 24, second paragraph, in the Discussion.

Results

Table 3, legend: here ‘loss of smell/taste’ is mentioned instead of ‘anosmis/ageusia’. Please be consistent (but also adjust according to the general comment above). Same comment for Table 4, legend.

Supplementary Table 1: Two columns ‘Initial test result’ and ‘Repeat test result’ seem superfluous, since for all HCW in this S1 Table the initial test results were negative and the ‘Repeat test results’ were positive, since this was the objective of this Table. I suggest removing these 2 columns and to provide a legend for this Table in which is indicated that for all HCW initial test results were negative and the repeat test results were positive.

-Page 19, last sentence of the Results: please replace ‘requiring’ by ‘required’ (twice), ‘needing’ by ‘needed’ and ‘dying’ by ‘died’.

Discussion

Page 19, last sentence: a total of 16% is mentioned, but in the abstract this is 14%. Probably the ‘false negatives’ are involved here. Please mention this more explicitly.

Reviewer #3: The authors present the study for HCWs with progressive symptoms and the SARS-CoV-2 test. Initial symptoms and their associations with SARS-CoV-2 RT PCR assay among 592 patients in 19 days were studied. Authors found 16% of HCW were diagnosed with clinical COVID-19, accruing a risk of 2% of their workforce.

The most significant symptoms associated with positive SARS-Cov-2 Assay remains spike in fever, with additional signs such as anosmia and myalgia. Specially HCW with anosmia/ageusia had higher odds (seven-fold) for having positive PCR, followed by a temperature of >-37.5 °C had 3.5-fold odds of having a positive PCR test.

In my opinion, the nine patients with a negative test (False Negative group) with symptom progression are the most exciting finding. As data from this group emphasizes the critical time point, where now the amount of viral load in the host is in the detection limit of PCR, and it also corresponds to showing symptoms in the host. Considering that the virus has 15 days of the incubation period, the critical question to be asked is, which is the most optimal time a health worker should go for testing? Or how often a health worker should get tested considering they show mild symptoms.

What titer of the virus from the host (after being symptomatic) is detectable by PCR?

Do symptoms and viral load depend on other independent variables like prior health issues/male/stress at work for HCW work.

I feel this study also emphasizes that measuring temp is vital for HCW, and such measures can be expanded across the hospitals and other institutes to limit the spread of the virus.

Comments :

How many groups were selected for fitting the Hosmer and Lemeshow test should be mentioned in the manuscript for clarity.

Finally, considering this unique collection of this HCW data, this dataset can be further explored by employing a predictive machine learning algorithm. A logistic regression model with R-square or may be softmax regression to predict the odds for the asymptotic individual or individual with one or more symptoms, to have a positive RT test, can be used.

6. PLOS authors have the option to publish the peer review history of their article (what does this mean?). If published, this will include your full peer review and any attached files.

Reviewer #1: No

Reviewer #2: Yes: Sylvia Bruisten

Reviewer #3: No

---

## [Author Response · Author response to Decision Letter 0]

14 Jun 2020

Journal Requirements/Editorial Comments:

Response: Thank you. We have revised the format/file names according to PLOS ONE’s requirements throughout the paper and associated files.

2. Please include additional information regarding the survey or questionnaire used in the study and ensure that you have provided sufficient details that others could replicate the analyses.

For instance, if you developed a questionnaire as part of this study and it is not under a copyright more restrictive than CC-BY, please include a copy, in both the original language and English, as Supporting Information.

Response: Thank you. We have attached the HCW triage form we used for telephonic triage as Supporting Information and mention this explicitly in our methods. 

S1 File. The healthcare system’s occupational health COVID-19/Influenza like illness triage form

Response to Reviewers:

Reviewer #1: 

1. Dr. Lan et al. has presented a retrospective observational study in which they investigated the presenting symptoms most predictive of positive/negative SARS-CoV-2 RT-PCR results among healthcare workers. Although the manuscript is well written and the methods are adequate, the observations made have already been described in the general population. HCWs should not be an exception.

Response: Thank you for your comments. Although the findings have been reported in the general population, SARS-CoV-2 testing was prioritized for HCWs per US CDC and other professional guidelines. During our study period, the HCWs in our healthcare system actually had a 6.7-fold propensity of being tested as compared to citizens of the Massachusetts general population. As a result, our population was considerably younger and tended to exhibit less symptoms than previous studies. In particular, according to the guidelines in place in March and April, asymptomatic persons were specifically blocked from testing. Second, to the best of our knowledge, previous observations in the general population were mostly based on cross-sectional studies that measured subjects’ symptoms and SARS-CoV-2 RT-PCR results at the same time or after receipt of test results, using self-reported surveys [1,2]. In our present study, however, we assessed HCWs’ symptoms at telephonic triage, which took place before they were referred to SARS-CoV-2 RT-PCR testing. Therefore, our symptom assessments were largely free of recall bias. Furthermore, the triage symptoms were validated by medical personnel for each HCW, and such validation had not been done in the research investigating the general population. Finally, focusing on HCWs helps to make decisions on not only prioritizing workers’ SARS-CoV-2 RT-PCR testing in healthcare settings, but also planning HCWs’ return-to-work, especially when the testing capacity is limited. Mildly symptomatic persons who were not HCWs were simply told to self-isolate.

References:

1. Menni C, Valdes A, Freydin MB, Ganesh S, El-Sayed Moustafa J, Visconti A, et al. Loss of smell and taste in combination with other symptoms is a strong predictor of COVID-19 infection. medRxiv. 2020:2020.04.05.20048421.

2. Menni C, Valdes AM, Freidin MB, Sudre CH, Nguyen LH, Drew DA, et al. Real-time tracking of self-reported symptoms to predict potential COVID-19. Nat Med. 2020.

Reviewer #2: 

1. This manuscript concerns a study among health care workers (HCW) and factors that are associated with testing positive for SARS-2 coronavirus. The data are sound and interesting and may be helpful to determine efficient testing strategies. Some points need to be considered and adjusted before this paper is suitable for publication in PlosOne.

Response: Thank you for this very positive assessment. Please find the point-by-point responses below to each specific comment.

General comments:

2. The Discussion section is rather long and may be shortened by removing redundant statements. For example page 22, second paragraph: this overlaps with other statements in the Discussion. 

Response: Thank you for your comments. We have removed that paragraph and streamlined other areas of the discussion. 

3. In the conclusions of the abstract and at the end of the manuscript, is stated that anosmia/ageusia and fever were the two strongest predictors of positive assays. In Table 1 and Table 4 however, it is clear that also myalgia is significantly associated with positive SARS-2 test results. This is also mentioned in the discussion. Why is this not mentioned in the conclusions in both abstract and discussion?

Response: Thank you for your suggestions. Indeed, myalgia is a significant predictor associated with SARS-CoV-2 PCR assays. We have added myalgia into the conclusions of both the abstract and the discussion. 

4. In addition: please mention as early as possible in the manuscript that ‘anosmia/ageusia’ means ‘smell/taste’ since these are medical terms with which not all readers are familiar.

Response: Thank you for your suggestions. We have added an explanation in the parenthesis after the first appearance of “anosmia/ageusia” appearing in both abstract and the manuscript body that these terms mean “lack/loss of smell and taste”, respectively. 

5. Could the authors speculate on the usefulness of their data for the general population? So would these indicators for a positive SARS-2 test also be true for non-HCW?

Response: Thank you for this suggestion. We have some text as our study strengths in the discussion to further elaborate the generalizability of our research: 

“… Finally, our present study included all eligible employees in the healthcare system. The study population comprised not only healthcare professionals but also staff/contractors such as maintenance or IT (Information Technology) persons. Therefore, our results may be generalized to other working populations during the pandemic.”

Specific comments:

Abstract

6. Page 3, 5th line: please insert ‘was’ between ’but’ and ‘associated’.

Response: Thank you for your suggestions. We have revised the sentence as per below:

“Anosmia/ageusia (i.e. loss of smell/loss of taste) was reported less frequently (16%) than other symptoms by HCWs with positive assays, but was associated with more than a seven-fold…”

Methods

7. Page 7/8: here the RT-PCR assays are mentioned and it is stated that results were either ‘positive’ or ‘negative’. Does this mean that there were no ‘indeterminate’ results? That is a bit hard to believe. If there were indeed ‘indeterminate’ results, how were these dealt with?

Response: Thank you for your comments. The laboratories used during the study period reported results only qualitatively as either “positive” or “negative”. In theory, a result can also return as “invalid” and a repeat sample is requested. This case is rare (~1 out of every 30,000 tests), and did not occur in this population. We added to the methods: “In case of an invalid or indeterminate result, a repeat sample is requested.”

8. Page 8, Data collection, second paragraph: please replace ‘Absence of a gold standard’ by ‘Because there is no reference diagnostic test’, since the term ‘gold standard’ is not appropriate. The same point for page 24, second paragraph, in the Discussion.

Response: Thank you for your suggestions. We have revised the sentences accordingly.

Results

9. Table 3, legend: here ‘loss of smell/taste’ is mentioned instead of ‘anosmia/ageusia’. Please be consistent (but also adjust according to the general comment above). Same comment for Table 4, legend.

Response: Thank you for your suggestions. We have revised ‘loss of smell/taste’ to ‘anosmia/ageusia (i.e. loss of smell/loss of taste)’ in the legends of both Table 3 and Table 4.

10. Supplementary Table 1: Two columns ‘Initial test result’ and ‘Repeat test result’ seem superfluous, since for all HCW in this S1 Table the initial test results were negative and the ‘Repeat test results’ were positive, since this was the objective of this Table. I suggest removing these 2 columns and to provide a legend for this Table in which is indicated that for all HCW initial test results were negative and the repeat test results were positive.

Response: Thank you for your suggestions. We have removed the two columns of S1 Table and revised its title as below.

“S1 Table. Presentation of HCWs with initial negative assays but repeat positive assays (i.e. presumed initial false negative assays)”

11. Page 19, last sentence of the Results: please replace ‘requiring’ by ‘required’ (twice), ‘needing’ by ‘needed’ and ‘dying’ by ‘died’.

Response: Thank you for your suggestions. We have revised the sentence as below.

“A total of nine among the 92 incident HCW COVID-19 cases (9.8%) (all nine diagnosed with the initial PCR assay) experienced a complication during the study period: four required an emergency room visit, five needed hospitalization, two of the hospitalized required intubation, and one of those died (1.1%).”

Discussion

12. Page 19, last sentence: a total of 16% is mentioned, but in the abstract this is 14%. Probably the ‘false negatives’ are involved here. Please mention this more explicitly.

Response: Thank you for your comment. We have revised the last sentence of Page 19 as below.

“We found that a total of 16% (including 14% with initial positive assays and 2% with initial false negative assays) of these HCWs were diagnosed with clinical COVID-19, yielding a cumulative attack rate of 2% for our workforce.”

Reviewer #3: 

1. The authors present the study for HCWs with progressive symptoms and the SARS-CoV-2 test. Initial symptoms and their associations with SARS-CoV-2 RT PCR assay among 592 patients in 19 days were studied. Authors found 16% of HCW were diagnosed with clinical COVID-19, accruing a risk of 2% of their workforce.

The most significant symptoms associated with positive SARS-Cov-2 Assay remains spike in fever, with additional signs such as anosmia and myalgia. Specially HCW with anosmia/ageusia had higher odds (seven-fold) for having positive PCR, followed by a temperature of >-37.5 °C had 3.5-fold odds of having a positive PCR test.

In my opinion, the nine patients with a negative test (False Negative group) with symptom progression are the most exciting finding. As data from this group emphasizes the critical time point, where now the amount of viral load in the host is in the detection limit of PCR, and it also corresponds to showing symptoms in the host. Considering that the virus has 15 days of the incubation period, the critical question to be asked is, which is the most optimal time a health worker should go for testing? Or how often a health worker should get tested considering they show mild symptoms.

Response: Thank you for your nice summary of our work and good questions. According to our study results and experience, a healthcare worker should undergo a SARS-CoV-2 testing when he/she develops systemic symptoms/signs (fever, body temperature ≥ 37.5°C, myalgia, and headache) and/or anosmia/ageusia. In contrast, if a healthcare worker has no symptoms or isolated sore throat/nasal congestion, he/she can usually forego testing, continue usual precautions and wait to see if they improve or worsen. Therefore, intensive self-monitoring of symptom progression would be crucial to determine when to go for a repeated test.

2. What titer of the virus from the host (after being symptomatic) is detectable by PCR?

Response: Thank you for this question. The limit of detection at the laboratory testing the majority of our samples is 100 copies of viral RNA/ml. We added a mention of this limit in the lab methods section.

3. Do symptoms and viral load depend on other independent variables like prior health issues/male/stress at work for HCW work?

Response: Thank you. We did not collect past medical history information or stress routinely. Regarding the HCWs’ age and sex, we included these variables in our models. Neither were significant predictors in univariate (Table 1) or in the multivariate models (Table 3, Table 4).

4. I feel this study also emphasizes that measuring temp is vital for HCW, and such measures can be expanded across the hospitals and other institutes to limit the spread of the virus.

Response: Thank you for your suggestions. We have modified the statement on temperature monitoring in the discussion to read as follows:

“… Our results support expert guidelines for temperature monitoring [18], which should not be limited to healthcare settings, and that a measured temperature ≥ 37.5°C may be predictive of a positive SARS-CoV-2 RT-PCR result. 

Comments :

5. How many groups were selected for fitting the Hosmer and Lemeshow test should be mentioned in the manuscript for clarity.

Response: Thank you for your comments. We have added selected group number of the Hosmer-Lemeshow goodness of fit test in “Methods” as below.

“…The Hosmer-Lemeshow goodness of fit test with 15 groups was also performed to check the model fit.”

6. Finally, considering this unique collection of this HCW data, this dataset can be further explored by employing a predictive machine learning algorithm. A logistic regression model with R-square or may be softmax regression to predict the odds for the asymptotic individual or individual with one or more symptoms, to have a positive RT test, can be used.

Response: Thank you for this suggestion. We agree it is an interesting approach. However, after some preliminary exploration with our limited sample, we concluded that it would be more fruitful to investigate this possibility with a much larger data set in the future.

---

## [Editor Report · Decision Letter 1]

17 Jun 2020

COVID-19 Symptoms Predictive of Healthcare Workers’ SARS-CoV-2 PCR Results

PONE-D-20-13207R1

Dear Dr. Kales,

We’re pleased to inform you that your manuscript has been judged scientifically suitable for publication and will be formally accepted for publication once it meets all outstanding technical requirements.

Kind regards,

Muhammad Adrish

Academic Editor

PLOS ONE

Additional Editor Comments (optional):

Thank you for revising the manuscript. You have satisfactorily answered all reviewer queries.
---

## [Editor Report · Acceptance letter]

19 Jun 2020

PONE-D-20-13207R1 

COVID-19 Symptoms Predictive of Healthcare Workers’ SARS-CoV-2 PCR Results 

Dear Dr. Kales:

I'm pleased to inform you that your manuscript has been deemed suitable for publication in PLOS ONE. Congratulations! Your manuscript is now with our production department. 

Kind regards, 

on behalf of

Dr. Muhammad Adrish 

Academic Editor

PLOS ONE